# Concept and Diagnostic Challenges of Renal-Limited Hemophagocytic Syndrome/Macrophage Activation Syndrome

**DOI:** 10.3390/jcm13082161

**Published:** 2024-04-09

**Authors:** Takahiro Uchida, Takashi Oda

**Affiliations:** Department of Nephrology and Blood Purification, Kidney Disease Center, Tokyo Medical University Hachioji Medical Center, Tokyo 193-0998, Japan; takashio@tokyo-med.ac.jp

**Keywords:** foamy glomerular lesion, glomerular lipidosis, hemophagocytic syndrome, histiocytic glomerulopathy, macrophage, macrophage activation syndrome

## Abstract

Hemophagocytic syndrome/macrophage activation syndrome (HPS/MAS) is a serious clinical condition that frequently leads to multiple organ failure, including acute kidney injury (AKI). Although the pathogenesis of AKI is not yet fully understood, it is believed to result from uncontrolled activation of the immune system involving macrophages and cytotoxic lymphocytes. Renal histology in HPS/MAS often presents with characteristic foamy glomerular lesions (glomerular lipidosis) with massive macrophage infiltration, known as histiocytic glomerulopathy. In this review, we introduce the recently proposed concept of renal-limited HPS/MAS as a novel etiology of histiocytic glomerular lipidosis. Patients with renal-limited HPS/MAS often develop AKI but do not fulfill the diagnostic criteria for HPS/MAS because their systemic manifestations are less severe. Therefore, the diagnosis largely depends on characteristic histological findings, that is, diffuse and global glomerular accumulation of foamy macrophages and cytotoxic lymphocytes accompanied by the interaction of these cells as well as the exclusion of various differential diseases. Although there are no established therapeutic regimens, these patients receive various types of therapies, including high-dose glucocorticoids, immunosuppressants, or anti-interleukin-1 drug, and generally achieve favorable outcomes. We summarized the concept, diagnostic challenges, and recent topics of this disease entity and discussed treatment options based on our own experiences.

## 1. Introduction

Hemophagocytic syndrome (HPS) is a life-threatening condition characterized by an uncontrolled activation of the immune systems [1,2]. In this condition, excessive activation of cytotoxic lymphocytes through some triggers drives proliferation and activation of macrophages, largely via the interferon–gamma (IFN-γ) pathway [3]. Macrophages, in turn, produce cytokines, such as tumor necrosis factor-alpha (TNF), interleukin-1 (IL-1), or IL-6, and further amplify immune responses, leading to a cytokine storm [4]. As a result, hemophagocytosis (the engulfment of self-hemocytes) and tissue/organ injury occur. There are two forms of HPS: a primary form that is associated with hereditary dysregulation of the immune system and mainly occurs in young children, and a secondary (acquired) form that occurs at any age and is associated with infection, malignancy, or autoimmune diseases [5].

The term macrophage activation syndrome (MAS) was first used by Stephan et al. in 1993 and was originally referred to as a variant of HPS that occurs as a complication of autoinflammatory or autoimmune diseases, such as systemic juvenile idiopathic arthritis in children and systemic lupus erythematosus or adult-onset Still’s disease in adults [6]. Indeed, hemophagocytosis accompanied by hematopoietic elements in the cytoplasm of macrophages was demonstrated in the initially reported cases of MAS [6]. However, thereafter, the histological presence of hemophagocytosis was found to be difficult to detect sometimes, particularly in the initial stages, both in HPS and MAS, necessitating repeat bone marrow biopsy [7]. In addition, features of hemophagocytosis are found in critically ill patients even without a diagnosis of HPS [5]. Thus, some classification criteria of HPS and MAS do not include the presence of hemophagocytosis as an essential diagnostic criterion [8,9]. In the absence of hemophagocytosis, the term HPS is somewhat disconcerting, and MAS seems more appropriate. Furthermore, the broad definition of HPS includes MAS as a cause of secondary HPS [8]. Thus, there is an overlap and confusion between the terms MAS and HPS, and it is difficult to distinguish their use. We therefore use the terms “MAS” and “HPS/MAS” synonymously in this article.

Patients with HPS/MAS often develop acute kidney injury (AKI), suggesting that the kidney is the target organ [10]. AKI itself causes high mortality and morbidity in the affected patients, and the occurrence of AKI in HPS/MAS is associated with adverse outcomes, including the development of chronic kidney disease [11]. Nonetheless, precise descriptions of the renal manifestations of HPS/MAS are scarce, partly because renal biopsies are seldom performed in these patients, especially during the acute phase. In recent limited reports on renal biopsy findings of HPS/MAS, the characteristic features of diffuse and global foamy glomerular lesions, namely glomerular lipidosis, have been presented. This characteristic pathology has been observed in various diseases but is generally divided into two major subtypes: glomerular lipidosis with extensive infiltration of CD68^+^ foam cells (histiocytes) and without histiocytic infiltration. With the accumulation of similar cases, the characteristic histological feature of glomerular lipidosis with histiocytic infiltration, termed histiocytic glomerulopathy, is now considered a typical renal lesion associated with HPS/MAS [12]. Thereafter, Roccatello et al. reported four patients with isolated renal involvement in HPS/MAS who presented with AKI and some features of HPS/MAS [4]. In these cases, the renal histological pattern uniformly presented as histiocytic glomerulopathy, and hemophagocytosis was found solely in the renal biopsy tissue. Recently, we reported a case that presented with similar pathological findings but milder clinical symptoms [13]. These cases are now considered potentially new disease entities and termed “renal-limited HPS/MAS” [4,13]. Systemic HPS/MAS is a clinical syndrome, and it is diagnosed through a series of systemic manifestations. In contrast, renal-limited HPS/MAS is a group of cases with ambiguous clinical symptoms that do not fulfill the diagnostic criteria, and diagnosis can be achieved only by suspecting the disease based on characteristic histopathological features.

This review presents an overview of our recent understanding of the renal pathological spectrum of HPS/MAS. We then introduce the detailed features of renal-limited HPS/MAS, including the diagnostic difficulties and dilemmas, to fill gaps in the current knowledge and increase awareness of it.

## 2. The Kidney as a Target Organ of HPS/MAS

Patients with HPS/MAS typically present with fever, hepatosplenomegaly, and lymphadenopathy. The laboratory findings include cytopenia, liver dysfunction, hyperferritinemia, hypertriglyceridemia, and hypofibrinogenemia [1].

In addition, patients with HPS/MAS often develop AKI with or without failure of multiple organs (MOF). Approximately 60–80% of patients with HPS/MAS admitted to the intensive care unit are reportedly suffering from AKI, and many of them require renal replacement therapy (RRT) [11,14]. Renal transplant recipients frequently develop HPS/MAS with renal involvement, and infectious triggers, such as cytomegalovirus, are common, possibly reflecting the immunosuppressive state of these patients [4]. Other infectious triggers are mostly viral, including Epstein–Barr virus and parvovirus B19 [15].

Although almost every structure of the kidney can be affected [10], acute tubular necrosis with or without tubulointerstitial inflammation, which consists of activated macrophages and cytotoxic lymphocytes, is the most common pathological finding [16,17]. Hemophagocytosis by infiltrating interstitial macrophages [18] and erythrophagocytic macrophages in the tubular lumen [19] have also been reported. Although glomerular involvement is less frequent, nephrotic syndrome can occur in the acute phase of HPS/MAS [11]. The reported renal pathologies are collapsing glomerulopathy (all of which are of African descent, suggesting the involvement of genetic factors), minimal change disease, and thrombotic microangiopathy (TMA) [18].

Glomerular lipidosis is presented not only by diseases involving genetic abnormalities in lipid metabolic pathways, such as lipoprotein glomerulopathy (LPG) or lecithin-cholesterol acyltransferase deficiency, but also those accompanied by intraglomerular infiltration of lipid-laden macrophages/histiocytes. However, the differences between these two disease entities can be subtle on light microscopy but become clear through immunostaining for CD68 or electron microscopy examination [12]. Mutations in apolipoprotein E (apoE) can result in histiocytic (apoE2 homozygote glomerulopathy) and non-histiocytic glomerulopathy (LPG), depending on the mutation types [20]. Recently, cases of HPS/MAS presenting with histological features of glomerular lipidosis with histiocytic infiltration have been reported; this lesion is now considered one of the typical renal lesions associated with HPS/MAS, and it is termed histiocytic glomerulopathy [7,12,15,21,22,23]. Mesangiolysis and injury of glomerular endothelial cells (swelling and loss of fenestrations of endothelial cells), histological features of TMA, may be observed in some of these cases. However, it should be noted that histiocytic glomerular lipidosis may also be observed in the conditions associated with clinical TMA [12].

## 3. The Concept and Characteristics of Renal-Limited HPS/MAS

The term renal-limited HPS/MAS was first used in 2022 as a potentially novel condition [4]. Although patients with renal-limited HPS/MAS present some features of HPS/MAS, they do not meet the current diagnostic criteria [5,8]. The renal histological pattern of the patients was uniform histiocytic glomerulopathy with the accumulation of numerous intraglomerular macrophages.

We recently reported a case of renal-limited MAS, which, to the best of our knowledge, was the first reported case of renal-limited MAS in a transplanted kidney [13]. A Japanese male in his forties, who underwent a living-donor kidney transplant for end-stage kidney disease (ESKD) due to an unknown original ailment, developed sub-nephrotic-range proteinuria (3.36 g/gCr) and increased serum triglyceride levels (500–600 mg/dL), indicative of type V hyperlipidemia, soon after the transplant. Although he received immunosuppressive therapy, consisting of tacrolimus, mycophenolate mofetil, everolimus, and glucocorticoids, serum levels of CRP and soluble IL-2 receptor were relatively elevated (1.13 mg/dL and 500 U/mL, respectively), suggesting an underlying inflammatory status. An allograft biopsy performed 6 months after the transplant showed glomerular lipidosis. Further histological analysis revealed an extensive glomerular accumulation of CD68^+^ lipid-laden macrophages and CD3^+^ cells (predominantly CD8^+^ cells) (Figure 1). Moreover, frequent contact between glomerular CD68^+^ and CD3^+^/CD8^+^ cells was shown by double immunoenzymatic or immunofluorescence staining, respectively, suggesting local and active interactions (Figure 2). Intensive lipid-lowering therapy was performed without modifying immunosuppressants, which reduced the glomerular lipid burden but did not decrease intraglomerular inflammation; the patient’s renal function further deteriorated thereafter.

In this case, a considerable number of CD57^+^ cells, some of which were CD8^+^, were observed in the glomeruli, and an increase in CD57^+^ T cells was also observed in the peripheral blood (Figure 3), whereas only a few CD56^+^ natural killer (NK) cells were observed. T cells expressing CD57, a NK cell marker, are innate immune cells that increase with age [24]. These cells reportedly collaborate with macrophages and play critical roles in in vitro models of the Shwartzman-reaction-like response [24]. The generalized Shwartzman reaction is a well-recognized experimental model of endotoxin shock or MOF and presumably occurs in humans [25], and the enhanced production of TNF from macrophages that is primed by IFN-γ produced by innate immune cells, especially CD57^+^ T cells, supposedly plays pivotal roles [26]. Decreased NK cell activity is one of the diagnostic criteria of HPS [8]; however, it cannot be measured in routine practice and is reportedly rarely helpful in the adult population [1]. On the other hand, the function of CD57^+^ T cells in HPS/MAS has not been thoroughly investigated. Hotta et al. reported the infiltration of CD57^+^ T cells in the glomeruli of older adult patients with membranoproliferative glomerulonephritis (MPGN) [27]; however, data regarding the involvement of these cells in glomerular diseases have been almost absent and must be further investigated.

Another patient in his forties demonstrated a histological pattern similar to that of previously reported cases of renal-limited MAS, namely intraglomerular accumulation of CD68^+^ macrophages and CD3^+^ T cells (Figure 4; unpublished observation). The patient initially showed nephrotic syndrome (urinary protein level, 8.0 g/day; estimated glomerular filtration rate, 47 mL/min/1.73 m^2^); however, the systemic manifestations of HPS/MAS were minimal. He was diagnosed with MPGN at that time and was treated with glucocorticoids and immunosuppressants. However, his renal function drastically worsened, which was triggered by an abdominal infection. He was pointed out as having multiple small abdominal visceral aneurysms and diagnosed with polyarteritis nodosa, and eventually led to ESKD requiring maintenance hemodialysis. Furthermore, a case of acute myelomonocytic leukemia that showed similar renal histological findings and later developed into systemic HPS/MAS was previously reported [28]. Thus, it is highly likely that more cases of renal-limited HPS/MAS remain undiagnosed. The characteristics of previously reported cases of renal-limited HPS/MAS are summarized in Table 1.

Recently, a case of HPS/MAS with isolated renal tubulointerstitial involvement was reported [29]. This report described a kidney transplant recipient who presented with AKI and features suggestive of HPS/MAS following multiple infections. The patient’s renal tissue showed massive macrophage infiltration solely in the interstitial areas, and the patient did not meet the current diagnostic criteria for HPS/MAS. The patient received antibiotics and antifungal and antiviral drugs, while the immunosuppressants were discontinued. She did not receive high-dose glucocorticoids or intravenous immunoglobulin and eventually evolved to graft loss requiring maintenance hemodialysis. Although further accumulation and analyses of such cases are needed, it could be cautiously suggested that renal-limited HPS/MAS is further divided into two subtypes: glomerular-limited and tubulointerstitial-limited.

Macrophages in mice are classified into two types based on their phenotype and function: F4/80^low^ CD11b^high^ monocyte-derived (bone marrow-derived) macrophages with proinflammatory cytokine-producing capacity (e.g., TNF or IL-12), and F4/80^high^ CD11b^low^ tissue-resident macrophages with phagocytic activity [30]. Recent studies have shown that the kidney contains both types of macrophages (Figure 5) [31,32] and that various disease conditions affect their function and proportion [31,33]. Thus, it seems reasonable that the activation of macrophages and their interaction with cytotoxic lymphocytes occur locally in the kidneys of patients with renal-limited HPS/MAS; however, future research is needed to evaluate this hypothesis.

## 4. Diagnosis of Renal-Limited HPS/MAS

Currently, there are no established diagnostic criteria for renal-limited HPS/MAS. Cases of renal-limited HPS/MAS do not fulfill the diagnostic criteria for systemic HPS/MAS and can be diagnosed solely based on the characteristic histological features of the renal tissue. In concrete terms, the key to diagnosis lies in the observation of diffuse and global glomerular accumulation of foamy macrophages and cytotoxic lymphocytes as well as their local interactions in cases with unexplained inflammatory findings. Furthermore, a thorough exclusion of various differential diseases is required. Thus, it is difficult to diagnose this disease.

## 5. Treatment Strategies for Systemic HPS/MAS and Renal-Limited HPS/MAS

Immunomodulatory therapy has shown marked improvements in survival in most etiologies of systemic HPS/MAS. Empiric administration of glucocorticoids, anti-IL-1 drugs, and/or intravenous immunoglobulin is now endorsed, depending on the severity of the patient’s condition, although no randomized controlled trials have been conducted [34]. The efficacy of intravenous cyclophosphamide has also been reported [35]. The need for supportive therapy, including transfusion of blood products, disseminated intravascular coagulation treatment, or RRT, should also be evaluated, and detailed and repeated investigations of the predisposing conditions or triggers of the patients are crucial.

Previously reported patients with renal-limited HPS/MAS have received various types of therapies, such as high-dose glucocorticoids, immunosuppressants, or anti-IL-1 drug, and generally achieve favorable outcomes (Table 1). However, there are no established therapeutic regimens, and future studies involving a larger number of participants are required. Lipid-lowering therapy was reportedly effective in LPG [36], which presents a similar renal histological pattern, but was ineffective in our patient with renal-limited MAS [13]. In this regard, hyperlipidemia (hypertriglyceridemia), observed in both LPG and HPS/MAS, is considered a direct cause of the renal lesions in the former but may be a consequence of disease pathogenesis in the latter. It could therefore be suggested that therapeutic strategies targeting hyperlipidemia alone are inadequate and inappropriate for patients with renal-limited HPS/MAS, even if the systemic manifestations are mild. A case of a kidney transplant recipient suggestive of renal tubulointerstitial-limited HPS/MAS experienced graft loss [29]. Although neither high-dose glucocorticoid treatment nor intravenous immunoglobulin was performed in the patient, they can be considered, especially if infections are under control. Thus, a common mistake made in the treatment of renal-limited HPS/MAS might be the avoidance of immunomodulatory therapy. Patients with renal-limited HPS/MAS, as discussed in this review, do not present with overt systemic manifestations but only manifest with glomerular lipidosis, which tends to make physicians hesitate to use immunomodulatory drugs. However, we would like to emphasize that the syndrome is primarily caused by the local activation of macrophages and their interaction with cytotoxic lymphocytes due to immunological abnormalities, and therefore immunomodulatory therapy is required.

## 6. Limitations

Because the concept of renal-limited HPS/MAS was just recently proposed, there are several issues to be solved. The first and biggest issue is that only a limited number of cases have been reported to date. In this regard, however, there seem to be more cases that remain undiagnosed. Our greater awareness is required for prompt recognition of this disease.

The secondary form of systemic HPS/MAS is associated with infection, malignancy, or autoimmune diseases, which seems to be applicable to the cases of renal-limited HPS/MAS from the literature research combined with our own experience (Table 1). However, future studies involving a larger number of patients that investigate the etiology of renal-limited HPS/MAS are needed. Although some overlap definitely exists between renal-limited and systemic HPS/MAS, longer follow-up of the patients will clarify the frequency of development from renal-limited to systemic HPS/MAS, which was observed in a previously reported case [28].

From the therapeutic point of view, previously reported patients with renal-limited HPS/MAS received various treatments, presumably depending on the individual physician’s judgement. There are no established therapeutic regimens yet, and clinical trials are desired in the future.

## 7. Concluding Remarks

Since the first description of HPS in the 1930s, the kidneys have been demonstrated to be one of the main target organs of HPS/MAS. Acute tubular necrosis, with or without tubulointerstitial inflammation, was originally considered the main pathology; however, later reports have shown that HPS/MAS affects almost every structure of the kidney. Podocytopathies (collapsing glomerulopathy and minimal change disease), TMA, and histiocytic glomerulopathy, which presents as glomerular lipidosis, are representative glomerular diseases. Importantly, both glomerular and non-glomerular diseases are frequently accompanied by AKI.

Recently, we and another group proposed the concept of renal-limited HPS/MAS [4,13]. The term renal-limited HPS/MAS, in the narrow sense, refers to the condition in which the characteristic feature of hemophagocytosis is detected only in the kidney but not in other organs, including the bone marrow. In contrast, cases in which hemophagocytosis is not observed in any organ, but signs of macrophage activation are observed locally in the kidney, can also be broadly categorized as renal-limited MAS. The former type reportedly presents with AKI and shows clinical features suggestive of HPS/MAS, such as fever or cytopenia, whereas the latter type seems to show heavy proteinuria. The prevalence of AKI in this group remains unclear. Both types do not fully meet the current classification criteria for HPS/MAS and are diagnosed solely based on renal biopsy findings, which uniformly show histiocytic glomerulopathy. It is also possible that renal-limited HPS/MAS can be further divided into two subtypes: glomerular-limited and tubulointerstitial-limited.

Although there remain several unsolved issues, such as prevalence, etiologies, and treatment strategies, greater awareness of this recently proposed disease is required for accurate diagnosis and therapeutic approaches. In this regard, electron microscopy examination is important for identifying the features of hemophagocytosis. However, it should be noted that detecting such findings is sometimes difficult in limited biopsy tissue samples, particularly in the early disease phase, and that electron microscopy is not available in some institutions. In contrast, noninvasive urine tests can be performed repeatedly and are considered to reflect the condition of the entire kidney [37]. The observation of phagocytic macrophages in urinary sediments could provide important diagnostic clues for renal HPS/MAS, as previously reported [19,38]. It should also be stressed that the histological analysis of glomerular infiltrating cells using various markers, including macrophages, cytotoxic lymphocytes, and activation status (such as HLA-DR), is crucial for the accurate recognition of pathogenesis.

## Figures and Tables

**Figure 1 jcm-13-02161-f001:**
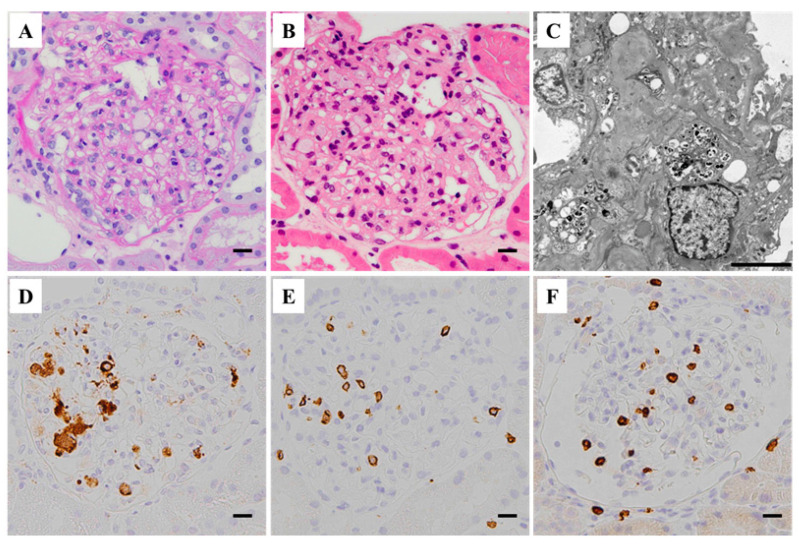
Histological features of the renal biopsy tissue performed 6 months after the transplantation in a patient with renal-limited hemophagocytic syndrome/macrophage activation syndrome, showing typical glomerular lipidosis (cited from ref. [13]). (**A**,**B**) Light microscopy images showing numerous foam cells within the glomerular capillaries (**A**, periodic acid Schiff reaction stain; **B**, hematoxylin and eosin stain; scale bars = 10.0 μm). (**C**) Electron microscopy image showing foam cells in a glomerulus. Scale bar = 5.0 μm. (**D**–**F**) Immunoperoxidase staining images showing extensive glomerular accumulation of vacuolated CD68^+^ cells (**D**), intermingled with CD3^+^ cells (**E**), most of which are positive for CD8 (**F**). Scale bars = 10.0 μm.

**Figure 2 jcm-13-02161-f002:**
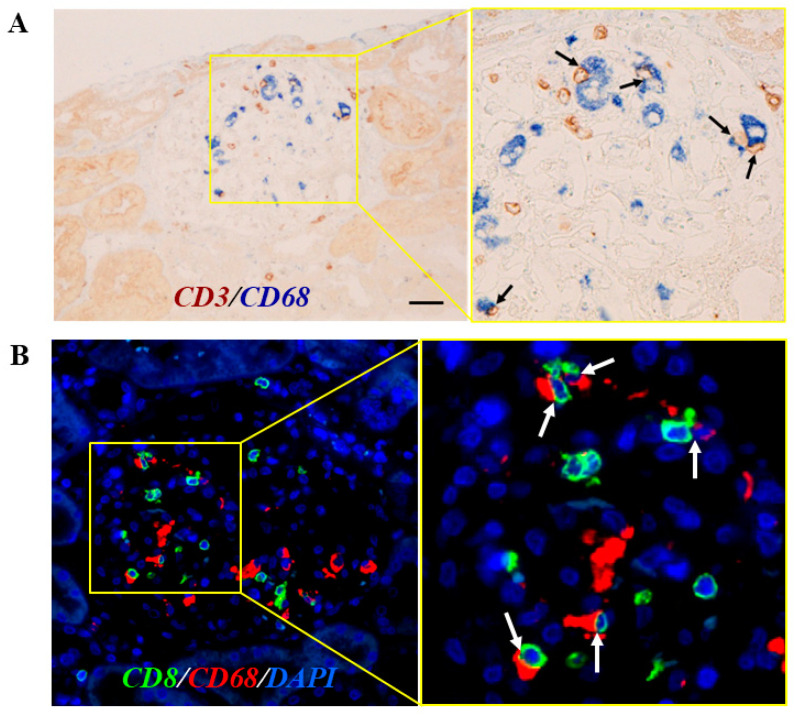
Frequent contact of macrophages and cytotoxic T cells in glomeruli of the same patient demonstrated in Figure 1. ((**A**), cited from ref. [13]) Double immunostaining for CD68 (alkaline phosphatase, blue) and CD3 (peroxidase, brown). Extensive glomerular accumulation of CD68^+^/CD3^+^ cells is shown (scale bar = 20.0 μm), and close contact between these cells is demonstrated (arrows) at a higher magnification. (**B**) Double immunofluorescence staining for CD68 (Alexa Fluor 594, red) and CD8 (Alexa Fluor 488, green) together with DAPI nuclear staining (blue) demonstrates the frequent contact of CD68^+^ and CD8^+^ cells within the glomerulus (arrows at a higher magnification). Scale bar = 10.0 μm.

**Figure 3 jcm-13-02161-f003:**
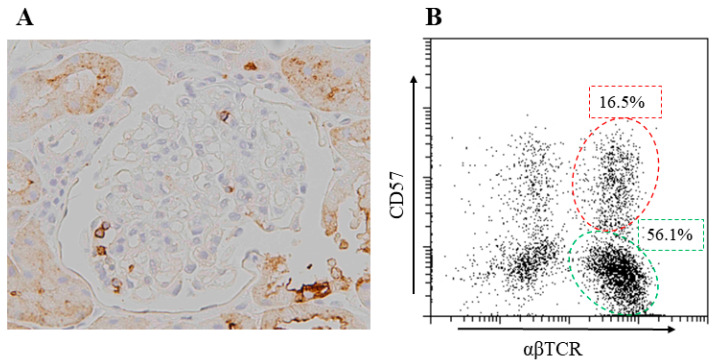
Intraglomerular infiltration of CD57^+^ T cells and their increase in the periphery of the same patient demonstrated in Figure 1 and Figure 2. (**A**) Light microscopy image showing glomerular infiltrating cells that are positive for CD57. (**B**) Flow cytometry analysis of peripheral blood mononuclear cells that are stained with anti-CD57 and anti-αβ T cell receptor (TCR) antibodies showing high proportion of CD57^+^ T cells (red dotted circle). αβTCR^+^ regular T cells are enclosed in green dotted circle. The number shows the proportion of each cell group.

**Figure 4 jcm-13-02161-f004:**
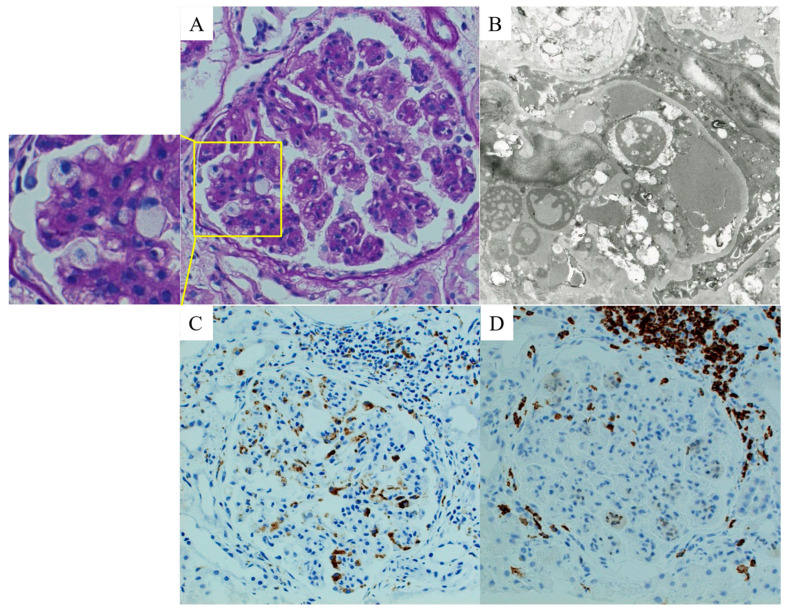
Representative renal pathological findings of a patient with renal-limited macrophage activation syndrome. (**A**) Light microscopy image showing a glomerulus with membranoproliferative glomerulonephritis pattern (periodic acid Schiff reaction stain). Multiple foam cells are also observed (inset). (**B**) Electron microscopy image showing lipid accumulation in a glomerular capillary wall. (**C**,**D**) Immunoperoxidase staining images showing numerous intraglomerular infiltrations of both (**C**) CD68^+^ macrophages and (**D**) CD3^+^ T cells.

**Figure 5 jcm-13-02161-f005:**
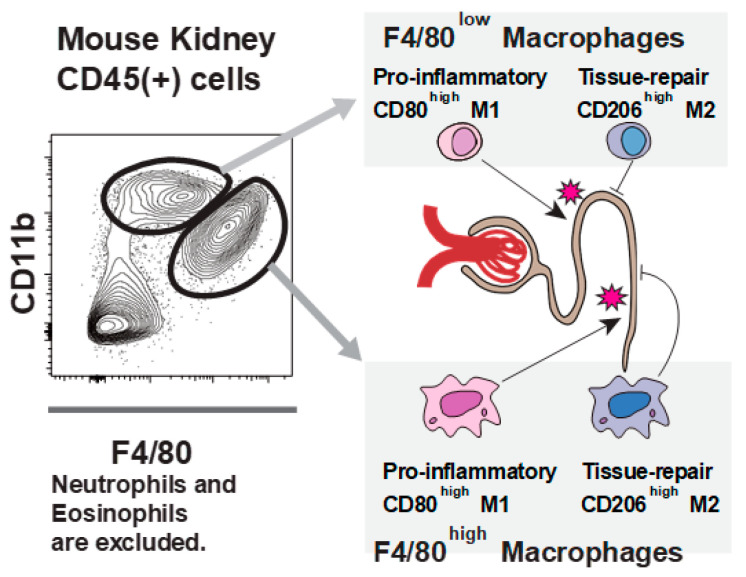
Two distinct macrophage phenotypes within the kidneys of mice (cited from ref. [32]). Two macrophage populations exist in the kidney: one with F4/80 high, and the other with low. Similar to other organs’ macrophage populations, such as liver Kupffer cells and pulmonary alveolar macrophages, F4/80 high cells are tissue-resident, and F4/80 low cells are monocyte-derived populations. Both types of macrophages express M1 (CD80) and M2 (CD206) markers. The balance between these antigens defines their function, that is, proinflammatory or tissue repair.

**Table 1 jcm-13-02161-t001:** Summary of the cases of renal-limited HPS/MAS.

Ref.	Age/Sex	Possible Trigger	DX of HPS/MAS *^1^	Renal Presentation	sCr	UP	Hemophagocytosis in Bone Marrow	Renal Pathology	Intraglomerular Hemophagocytosis	Treatments	Kidney Recovery
[22]	45/F	Ovarian cancer	No	AKI	1.9	10.5	N.D.	HG	No	Chemotherapy	Yes
[4]	83/M	Pneumonia	No	Anuric AKI	10.1	N.D.	No	HG with extracapillary lesion	Yes	Glucocorticoids	Yes
[4]	79/M	Airway infection	No	Anuric AKI	4.7	N.D.	No	HG with FSGS	Yes	Glucocorticoids	Yes
[4]	69/M	Airway infection	No	Recurrent AKI	3.7	4.8	No	HG	Yes	Canakinumab	Yes
[4]	70/F	Pneumonia	No	Deterioration of RF	2	2.4	No	HG	Yes	Anakinra	Yes
[13]	42/M	Unknown	No	Proteinuria	1.1	3.8	N.T.	HG	No	Pemafibrate *^2^	No
[28]	59/M	Unknown	No	NS	1.4	>10	No *^3^	HG with MPGN pattern	N.D.	Glucocorticoids	Yes
- *^4^	47/M	Unknown	No	NS	1.3	8	N.T.	HG with MPGN pattern	No	Glucocorticoids, CsA	No

AKI, acute kidney injury; CsA, cyclosporine; DX, diagnosis; FSGS, focal and segmental glomerulosclerosis; HG, histiocytic glomerulopathy; HPS, hemophagocytic syndrome; MAS, macrophage activation syndrome; MPGN, membranoproliferative glomerulonephritis; NS, nephrotic syndrome; N.D., not described; N.T., not tested; RF, renal function; sCr, serum creatinine (mg/dL); UP, proteinuria (g/day). *^1^ Based on the HLH-2004 criteria [8]. *^2^ The patient was on maintenance immunosuppressive therapy after kidney transplantation. *^3^ The patient thereafter developed systemic HPS/MAS and died. *^4^ The patient was later diagnosed with polyarteritis nodosa (unpublished observation).

## Data Availability

All datasets generated for this study are included in the article. Further inquiries can be directed to the corresponding author.

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
