# Peer review of "Concept and Diagnostic Challenges of Renal-Limited Hemophagocytic Syndrome/Macrophage Activation Syndrome"

_jcm, 2024, doi:10.3390/jcm13082161_

Round 1

Reviewer 1 Report

Comments and Suggestions for Authors

This is an up-to-date as well as comprehensive review on renal limited HPS/MAS.

I would like the authors to improve a few points below.

1.       Overall: recently, ‘glucocorticoids’ is considered as more appropriate word than steroids. Please consider.

2.       In introduction, the authors stated that infection, malignancy and autoimmune disease are included as causes of secondary HPS/MAS. In contrast, no cases of renal limited HPS/MAS secondary to autoimmune disease are included in Table 1. Are there any reports of renal limited HPS/MAS secondary to autoimmune disease? Or can we exclude autoimmune disease as causes of secondary renal limited HPS/MAS? Please discuss it.

3.       Line 180: 2 in ‘m2’ to denote squaremeter should be superscript

Reviewer 2 Report

Comments and Suggestions for Authors

The manuscript does a good job of summarizing the main points of the review, including the condition's pathogenesis, diagnostic challenges, and treatment options. It mentions the novel concept of renal-limited HPS/MAS, the importance of renal histology for diagnosis, and touches upon treatment based on the authors' experiences. To enhance completeness, consider whether the abstract briefly mentions the outcomes or effectiveness of the discussed treatment options or the implications of this novel etiology on patient management and prognosis. The manuscript indicates a significant contribution to the field by introducing a novel etiology for histiocytic glomerular lipidosis and highlighting the diagnostic challenges of renal-limited HPS/MAS. It suggests that this review could help clinicians recognize and diagnose a subset of patients who might otherwise be overlooked. To strengthen this section, the manuscript could briefly mention how this review fills gaps in current knowledge or changes clinical practice.

Reviewer 3 Report

Comments and Suggestions for Authors

The manuscript <Concept and Diagnostic Challenges of Renal-limited Hemoph-  phagocytic Syndrome/Macrophage Activation Syndrome> by Takahiro Uchida and Takashi Oda is a review of the actual understanding of the renal pathological spectrum of HPS/MAS and renal-limited HPS/MAS, including the diagnostic difficulties and dilemmas. The paper consists of 6 parts the introduction, a part in which the kidney is described as an affected organ, the characteristics of the disease that is limited to the kidney, diagnosis and treatment at the end of the kidney. The work is richly illustrated and documented tabularly and graphically. references are listed according to journal rules. the work is a valuable contribution to a syndrome that has been little studied so far.

Round 2

Reviewer 1 Report

Comments and Suggestions for Authors

(No further comments)